# Investigating Factors Associated with Migration and Cultural Adaptation in Relation to Change in Attitudes and Behavior towards Female Genital Mutilation/Cutting (FGM/C) among Populations from FGM/C-Practicing Countries Living in Western Countries: A Scoping Review

**DOI:** 10.3390/ijerph21050528

**Published:** 2024-04-24

**Authors:** Nasteha Salah, Sara Cottler-Casanova, Patrick Petignat, Jasmine Abdulcadir

**Affiliations:** 1Division of Gynaecology, Department of Paediatrics, Gynaecology and Obstetrics, Geneva University Hospitals, 1211 Geneva, Switzerland; saraalison.cottler-casanova@hcuge.ch (S.C.-C.); patrick.petignat@hcuge.ch (P.P.); 2Institute of Global Health, University of Geneva, 1202 Geneva, Switzerland

**Keywords:** female genital mutilation, female genital cutting, cultural change, attitude, Western countries, abandonment of FGM/C, FGM, FGC

## Abstract

A growing body of evidence indicates a significant decrease in support for female genital mutilation/cutting (FGM/C) within post-migration communities in Western countries. Addressing knowledge gaps and comprehending the factors associated with FGM/C discontinuation in these communities is crucial. The objective of this scoping review is to describe the effects of migration and cultural change on factors supporting FGM/C cessation in migrant communities. The review, from 2012 to 2023, included the following databases: Embase, PubMed, Google Scholar, Swisscovery, CINAHL, APA PsycInfo, and gray literature. Applying the PRISMA-ScR framework, we identified 2819 studies, with 17 meeting the inclusion criteria. The results revealed seven key factors shaping attitudes and behavior toward FGM/C abandonment: (1) legislation against FGM/C, (2) knowing that FGM/C is not a religious requirement, (3) enhancing education about the practice, (4) migration and cultural change, (5) awareness of the harmful effects of FGM/C, (6) a positive view of uncut girls, and (7) a sense of self-agency. These findings highlight factors on a social, community, interpersonal, and personal level that enhance the abandonment of the practice. Further research in the FGM/C field will gain more accuracy in understanding and accounting for these multilevel factors in post-migration settings, offering valuable insights for targeted interventions to promote the cessation of the practice.

## 1. Introduction

The term female genital mutilation/cutting (FGM/C) encompasses all non-medical interventions that totally or partially alter or harm the external female genitalia [1]. The World Health Organization defines four types of female genital mutilation/cutting (FGM/C): (Type 1) cutting of the clitoral hood or the clitoris; (Type 2) excision—cutting of the inner labia, with or without excision of the outer labia and/or the clitoris; (Type 3) infibulation—narrowing of the vaginal opening with apposition of the inner and/or the outer labia with or without excision of the clitoris; and (Type 4) all other harmful procedures on female genitalia for non-therapeutic purposes, such as piercing, incision, scarification, and cauterization [1]. FGM/C practices can have serious short- and long-term consequences, with physical, psychological, obstetric, and psychosexual complications that can affect the quality of life of women and girls [2]. The practice is prevalent in Africa, the Middle East, and Asia [1]. UNICEF estimates that least 230 million women and girls from 31 countries have undergone some form of female genital mutilation/cutting [3]. Due to global migration, FGM/C is not limited to traditionally practicing countries. The European Network End FGM estimates that around 600,000 women and girls with FGM/C live in Europe [4]. Research indicates that migration and cultural adaptation significantly influence attitudes toward FGM/C [5].

The concept of cultural change refers to the process through which migrants integrate into and become part of a new society. It involves the gradual adaptation of social, psychological, and cultural aspects as ethnic groups align with societal norms over time [6]. Immigrants may undergo a process of abandoning certain cultural practices that carry high social and legal costs as part of an adaptive strategy. The first generation of migrants may gradually leave behind old cultural customs, such as FGM/C, and adopt other values and norms of the host community. However, social adaptation can be jeopardized by social barriers such as racism and unequal power structures in the host country, discrimination, and other inherent barriers associated with migration, such as economic and social precarities [7,8]. In the migration context, for populations coming from countries where FGM/C is traditional, factors measuring adaptation to the new culture traditionally include language proficiency, age at arrival, length of residence in the host country, cross-border social networks, and social interactions. This intricate process involves integrating into the social networks of the host country, influenced by various factors such as family, cultural background, social class, work situation, and other relevant aspects [9]. Cultural adaption may occur at different dimensions and rates, interacting with other processes. The ecological models of health behavior emphasize these modifications of individual behaviors while considering the impacts of social, physical, and political environments. The analysis spans multiple levels, encompassing social, community, interpersonal, and personal factors that influence behavior. This approach provides insight into the ways in which diverse factors contribute to shaping attitudes and norms related to the cessation of FGM/C practices [10]. The migration context appears to be a favorable platform for the change of views towards the practice of FGM/C. Research indicates that the perspective of young individuals in the second generation in Britain is that FGM/C is a historical phenomenon detached from their own experiences [7]. Post-migration cultural change can influence the risk estimation on girls of FGM/C post-migration [11]. Biased data can perpetuate a misleading representation of the phenomenon among the general public, the media, healthcare professionals, social workers, law enforcement, and the political sphere, sometimes resulting in counterproductive effects in the prevention of FGM/C within communities [12]. For stakeholders, researchers, and experts engaged with populations originating from FGM/C communities, understanding the factors associated with cultural shifts is important. Cultural adaptation involves a complex process influenced by interconnected dimensions, subsequently impacting the health and well-being of individuals if not properly addressed or if stigmatized. Comprehending the composition of these communities, including intergenerational differences and their perspectives and beliefs regarding the practice, is essential for broadening the scope of policies to encompass social, environmental, and behavioral dimensions relevant to attitudinal change. This review represents the preliminary stage of broader studies conducted as part of a PhD project. The scoping review objective is to describe factors related to the effects of migration and cultural change that support the discontinuation of FGM/C among post-migrant populations coming from FGM/C-practicing countries and living in Western countries.

## 2. Materials and Methods

This is a scoping review with the following inclusion and exclusion criteria.

### 2.1. Inclusion Criteria

Literature that describes factors that influence change in attitude towards the discontinuation of FGM/C among populations originated from FGM/C-practicing countries living in various Europe countries, North America, Canada, Australia, New Zealand, and Israel.Articles comparing attitude changes within the same population between those who remained in their country of origin and those who migrated were included in the analysis.Articles comparing factors perpetuating or hindering FGM/C in post-migration populations living in Western countries.Original research articles spanning from 2012 to 2023.Studies focused on factors associated with attitudes and behavior change regarding FGM/C among migrants above 16 years old from countries where FGM/C is prevalent, now living in the West.Literature conducted in the English and French languages.

### 2.2. Exclusion Criteria

Studies predating 2012.Studies related to the perception of healthcare professionals and their knowledge and attitude towards FGM/C.Studies not related to factors influencing attitude changes towards FGM/C in post-migration populations living in Western countries (literature looking mainly to factors that perpetuate the practice).Studies related to change in attitudes and behavior towards FGM/C in populations living in FGM/C countries.Non-original research articles such as reviews.Research conducted in languages other than English and French.Studies not available in full text.Non-relevant or unrelated research that does not address the stated aim.

### 2.3. Search Strategy

The scoping review, conducted on 16 June 2023, targeted publications indexed in the following databases chosen based on their relevance as sources for medical, behavioral, and healthcare research: Embase, PubMed/Medline, Google Scholar, Swisscovery, CINAHL, and APA’s PsycInfo. We hand-searched the WHO, UNICEF, UNFPA, EndFGM Europe, and EIGE Europa websites, as well as gray literature including research papers and doctoral theses, in both French and English. The search terms were selected based on the terminology used in the National Library of Medicine’s MeSH for the PubMed platform, Embase PICO searches, and APA Psycho’s INDEX TERM.

The search terms were divided into four categories in Table 1, with specific keywords for each category linked together by Boolean operators such as AND and OR. Additionally, citations within relevant articles were searched to supplement the documentation.

### 2.4. Selection Strategy

Publications were downloaded from various databases and directly imported into Rayyan, a web application for conducting systematic reviews [13]. After duplication removal, the first author (NS) screened the titles and abstracts of all remaining publications for relevance. One co-author (SC) then screened the titles and abstracts of the selected publications (*n* = 75) to determine whether studies qualified for full-text review. Both reviewers read and evaluated all remaining full-text manuscripts included in the review, and any disagreements were resolved through discussion between the two reviewers based on inclusion and exclusion criteria.

## 3. Results

### 3.1. Synthesis of Results

The study selection process and results are presented in a PRISMA flow diagram Figure 1 [14]. The search strategy yielded a total of 2819 articles from five databases and one web search engine. A total of 980 duplicate records were automatically detected by the Rayyan program. Following the screening of titles and abstracts, 1754 articles were excluded, leaving 85. Out of these 85 studies, 10 were not accessible in full text. The selection included fifty-two peer-reviewed articles, two dissertations, three commentary articles, two systematic reviews, one scoping review, four review articles, two computerized tests, two books and one book chapter, one conference publication, one editorial, one newspaper, two case studies and one census. The authors read a total of 75 full-text records. Out of the selected articles, a total of 58 records were excluded based on the following reasons: 41 articles were not focused on factors that influence changes in attitude towards FGM/C. Two did not offer adequate clarity regarding the interconnection of factors associated with changes in attitudes towards FGM/C and migration. Eight articles were published before 2012, four did not investigate the intended population and three were not original research articles. The remaining 17 studies that matched the inclusion criteria were selected for the scoping review. There were ten qualitative studies (59%), six quantitative studies (35%), and one computerized test (6%). 

### 3.2. Quality Assessment

The researchers used the Critical Appraisal Skills Program (CASP) checklist for qualitative studies to assess the quality of the papers [15]. This checklist and scoring system was adapted for this review. Out of a total 10 questions, each question received a score of two for “Yes”, one for “Can’t Tell”, and zero for “No”. The total score (out of 20 possible) categorizes papers as low (0–10), moderate (11–13), or high quality (14–20). For quantitative observational cross-sectional studies, the Appraisal tool for Cross-Sectional Studies (AXIS) checklist was employed, with 20 questions scored similarly, resulting in categorization as low (0–19), moderate (20–30), or high quality (31–40), based on the overall score [16]. Sixteen studies were classified as being of high methodological quality, and the remaining one was deemed to be of moderate quality; none were rated as poor. Among the qualitative studies, excluding two [17,18], there was a consistent failure to adequately address considerations such as the relationship between the research and participants or justification of the research design [19], utilization of qualitative methodology to achieve research goals, or ethical considerations regarding obtaining consent [20]. In the case of quantitative studies, while information on measures taken to address non-responders was generally missing, one study lacked clear explanations on how and what type of consent was obtained [21]. Some studies did not clarify the existence of potential conflicts [22,23]. The moderate study displayed a combination of these shortcomings and a lack of rigor in the methodology [24].

### 3.3. Study Characteristics

Study publication dates ranged from 2012 [17] to 2022, with eight studies (47%) published after 2020. These studies varied in sample sizes, ranging from 12 [25] to 2344 [26], although most of the studies had relatively small sample sizes (70.6%) of fewer than 100 participants.

Studies were conducted in various Western countries, such as Norway [17,20,21,27], The Netherlands [28], Spain [18,25], Italy [29], Sweden [22,23], Belgium [30], Australia [31,32], the United States [19,24], and Canada [33]. One study took place in both Switzerland and Sudan [26].

The majority of the study populations were both male and female; five studies were only with female participants [18,20,25,30,33] and two studies included only male participants [29,31]. One study focused on young adults of 16 to 22 years old [27], while all remaining studies included participants who were 18 years old and above. Most participants predominantly originated from communities from East or West African countries [18,19,20,24,25,26,28,29,30,31,32,33], and five studies included solely participants originating from Somalia [17,21,22,23,27].

The two qualitative studies employed interviews, two studies used focus groups, one study combined semi-structured interviews and focus groups, one study utilized a combination of in-depth unstructured and semi-structured interviews, and one used a multimethod approach. Additionally, two studies adopted semi-structured interviews, while the two studies employed life stories through an open interview format. In the case of the six quantitative studies, a questionnaire served as the data collection tool for most of them. However, two studies employed an audio-computer-assisted self-interview (ACASI), where participants listened to audio instructions through headphones [24]. The second study by Vogt et al. required participants to answer using only three keys on a keyboard during the test [26].

### 3.4. Study Findings

The characteristics of the selected studies (N = 17), including type, country, study quality appraisal, study population, and aim, along with the factors identified within the article as factors associated with change in attitudes and behavior towards FGM/C, are reported in Table 2. We employed the Prisma for Scoping Reviews framework to analyze the results [14].

The review’s findings show that diverse factors play a role in shifting attitudes toward the abandonment of FGM/C among populations originating from countries where FGM/C is prevalent and who are residing in Western nations. Seven distinct factors were commonly identified in the selected studies, along with their sub-categories, and are shown in Table 3.

## 4. Discussion

This scoping review is part of a growing literature exploring factors influencing attitude shifts toward FGM/C among migrating populations in Western countries from regions where FGM/C is prevalent. Most participants in the reviewed studies expressed opposition to FGM/C in Western countries [17,19,21,23,24,26,27,28,29,30,31,32,33]. The results emphasize that factors contributing to attitude and behavior shifts regarding FGM/C can be explained by seven prominent factors linked with the sub-categories described below.

### 4.1. Legislation and Law against FGM/C in Host Countries and Legal Repercussions

In our analysis, we found that the primary factor associated with changes in attitudes toward FGM/C is the legal prohibition of FGM/C and the fear of encountering significant legal consequences in Western countries. Both quantitative and qualitative studies corroborated the deterrent effect against the practice [18,25,26]. In Koukoui et al.’s 2017 study conducted with mothers raising daughters who had not undergone FGM/C in Canada, there was a notable concern that their daughters might be removed by Youth Protection if FGM/C were performed on them [33]. Several studies, including those conducted in Norway, Australia, and Spain, reported that participants expressed no intention to subject their daughters to circumcision, primarily due to legal prohibitions or incarceration in their host countries [20,25,32]. Studies in the UK and Johnsdotter’s review across Europe noted that fewer than 50 criminal cases were identified, in comparison to the presence of hundreds of thousands of people from FGM/C-practicing countries in these regions [5,34]. Hodes et al.’s study in the UK and Ireland also found very few cases identified by pediatricians [35]. The knowledge of legal consequences and the possibility of refraining from the practice seemed to favor its abandonment [21]. The sub-category associated with this factor was identified as societal disapproval of FGM/C in Western countries [23]. Gele et al. identified the fact that the abandonment of FGM/C was attributed to a supportive anti-FGM/C environment in Norway among Somali immigrants, in contrast to the social pressure that perpetuate the practice in Somalia [17]. Enforcing legislation against FGM/C protects girls but may lead to discrimination based on ethnicity rather than actual circumstances. To mitigate this, action is necessary to minimize stigmatization within communities and improve risk assessment based on factors other than ethnic background. This involves increasing clear and relevant knowledge about FGM/C among professionals, authorities, police, and the media. For example, the ‘détectomètre’, a tool developed by GAMS in Belgium for preventing FGM/C, is also designed to support various social, health, or parental actors in providing assistance, care, and information related to FGM/C [36]. The legislation should not perceived as a mere imposition but also address protection and support [18]. Likewise, the UK study in Bristol highlighted the fact that local educational initiatives in communities with FGM/C have effectively contributed to diminishing the practice and to shifting attitudes [37].

### 4.2. FGM/C Is Not Mandated by Religion; It Is a Manifestation of Traditional Culture Rather Than a Religious Obligation

The second factor contributing to shifting attitudes towards FGM/C was the recognition that the practice was not religiously mandated, as found in the study by Gele et al. A total of 38 out of 39 participants expressed the belief that FGM/C was not a religious obligation, viewing this aspect as a significant reason for their decision to reject the tradition [17]. A majority of participants from Somalia living in the Netherlands maintained that it is inaccurate to claim that FGM/C is ‘Sunna’ in Islam, clarifying that only male circumcision holds the status of ‘Sunna’ in Islam [28]. Furthermore, participants living in Spain reported that FGM/C is a cultural practice rather than a religious one [18,25]. Berg et al.’s study suggested that religion might play a role in both perpetuating and discontinuing FGM/C [38]. To address this aspect, a Norwegian study, which examined individuals supporting FGM/C, highlighted the fact that a key moment in attitude change was when a religious leader declared that FGM/C was not a religious mandate [20], emphasizing the importance of raising awareness among the public and actively involving religious authorities. Establishing alliances with religious leaders who are against FGM/C and who often serve as normative figures, becomes crucial in discounting the practice [38,39].

### 4.3. Enhancing Education Regarding the Harm of FGM/C and Increased Knowledge of Human Anatomy

The third factor was enhancing education regarding the harm of FGM/C and increased knowledge of human anatomy. The 2016 study from Catania et al. on awareness of the health risks posed to women has emerged as the primary reason for rejecting the practice among 50 sub-Saharan African men. Information campaigns in Italy have significantly improved men’s understanding of the risks associated with FGM/C. Some participants reported becoming aware of these dangers only after migrating to Italy [29]. Similar patterns are observed among male migrants with university education residing in Australia, indicating that migration has transformed their prior beliefs and perceptions of FGM/C [31]. Within two sub-categories, community education and awareness raising have the potential to effectively decrease the prevalence of FGM/C [24]. Educating individuals about the health risks and detrimental effects linked to FGM/C significantly correlates with altering viewpoints across genders [25,27]. In the Agboli et al. study, education on anatomy and sexuality had a notable impact. A medical student noted that learning about physiologic vulvar anatomy and the consequences of FGM/C during anatomy lectures was enlightening and influenced a change in views [30]. Interactions with health professionals also led participants to reconsider their perceptions of a vulva with FGM/C, deepening their understanding of physical changes and negative health consequences linked to FGM/C [20]. The study by Lien et al. activists changed their perception because of four significant events. These included obtaining information from various educational institutions, attending seminars and conferences, working as interpreters in hospitals, and engaging in discussions with families and friends who opposed the practice. The authors concluded that the combined influence of information from diverse sources likely played a complementary and cumulative role in contributing to a shift in attitudes [20].

### 4.4. Effect of Migration and Cultural Change

The fourth factor, related to the influence of migration and cultural change, shows consistent trends in attitude change toward FGM/C over time. Within this factor, the first sub-category indicates that a longer duration of residence is associated with decreased support for FGM/C [22,31]. Individuals residing for less than five years were more likely to agree that FGM/C should be allowed under Australian law, while those with over five years of residence showed significantly lower agreement. Similarly, the desire for their partners to undergo FGM/C declined by 30% over five years, indicating a reduction in attitudinal support for FGM/C [31]. A study in Sweden comparing Somalis with 15 or more years of residency to those residing for less than 2 years revealed that the latter group had significantly higher odds of considering any form of FMG/C acceptable (11 times higher). This group also showed increased support for cutting their daughters and continuing FMG/C, with these associations remaining significant even after adjusting for other background factors [23]. Both studies consistently showed that the longer individuals reside in the host country, the less likely they are to support FGM/C [21,23,29]. Furthermore, the finding that 90% of girls arriving in Norway at age seven or younger were uncircumcised indicates the abandonment of FGM/C among Somali immigrants in Oslo even before migration [21]. The second sub-category indicated that reduced social pressure in the post-migration context favors less support for FGM/C, as highlighted in studies conducted in the Netherlands and Norway [17,28]. Gele and Johnsdotter suggested that migrants experience a shift in attitudes due to decreased traditional social pressures supporting the practice. This change is connected to less pressure from extended family members, influenced by geographical distance [21,40]. Relocating to a new country can alleviate cultural pressure to engage in FGM/C by facilitating the development of critical perspectives on the merits of adhering to the practice [9].

### 4.5. Awareness of the Negative Health Consequences of FGM/C

In our analysis, we identified that the fifth factor associated with attitudinal changes towards FGM/C was awareness of the adverse health effects of FGM/C. This factor was associated with two sub-categories: (1) increased concerns about sexual and reproductive health issues related to FGM/C, and (2) a shift in parental roles aimed at protecting daughters from FGM/C. Research highlights the fact that women and girls with FGM/C can have immediate or lasting consequences, such as discomfort during intercourse, painful urination, menstrual issues, infection risks, prolonged childbirth, and reduced sexual desire, compared to those without genital cutting [24]. Citing similar outcomes, Gele et al. found a strong association between discontinuing FGM/C and health concerns [27]. Shahawa et al. confirmed that participants’ convictions against FGM/C, especially in its severe form, stemmed from the acknowledged harmful effects. Opposition to the practice was fueled by the painful experiences of loved ones, prompting questioning of the validity of justifications and the continuation of the practice [19]. Siles-González’s study on women who have emerged as advocates against FGM/C, engaging with groups from diverse cultures, found that these women became aware of the abnormal negative health consequences of FGM/C. They recognized that some sexual and reproductive issues are not typical among women from other cultures who have not undergone FGM/C. Greater awareness and understanding of the adverse health consequences of FGM/C prompt individuals to question the significance of the practice, fostering a broader shift towards discontinuation [27]. The second sub-category is linked to parents’ motivation to protect their daughters from FGM/C, as seen in Canadian mothers who expressed a desire to discontinue the practice to prevent negative health consequences based on personal experience [33]. In a study in Spain, mothers took special precautions during travels to their countries of origin, fearing relatives might subject the girls to mutilation without their consent [25]. Migration increases awareness and shifts perceptions of FGM/C, emphasizing its negative health consequences and the duty to protect daughters from this practice.

### 4.6. Change in Perception towards Uncircumcised Girls and Better or Same Perspective of Marriage for Uncut Girls

In Gele et al.’s study, participants agreed that uncircumcised young girls were considered healthier and more attractive for relationships and marriage than their circumcised counterparts [17,21]. Another study by Gele et al. exploring attitudes among Somali-Norwegian youth found that being uncircumcised was considered the norm, and FGM/C was not linked to social status [27]. In the Akinsulure et al. study, male participants had no specific preference for dating or marrying uncut women, but those who did generally preferred women without FMG/C [24]. This finding aligns with other studies suggesting that undergoing FGM/C is no longer mandatory for marriage [26,28,40,41]. Additional research, including the Gutiérrez-García et al. study, indicates that both diaspora girls who experienced FGM/C before migrating and those born in Spain without FGM/C have similar marriage rates [41].

### 4.7. Individual Disposition to Oppose FGM/C

In our study, the seventh and final factor identified was linked to perceived behavioral control. Kawous et al.’s research found that newly arrived individuals in the Netherlands, especially those from Somalia and Eritrea, strongly opposed FGM/C for their daughters. They expressed the ability to resist social pressure by explaining their decisions, prompted by personal experiences, leading to a perception of injustice and a change in views of FGM/C. Higher social support and lower pressure correlated with greater perceived behavioral control, reinforcing the intention to abstain from FGM/C [28]. In the Norwegian study, a woman who was initially surprised to meet uncircumcised Muslim women in an asylum center, later, during nursing college, gained an understanding of her FGM/C status. The acquisition of new information in this context resulted in psychological pain for her, transitioning from personal pride in having FGM/C to feeling victimized. However, another woman in the same study found comfort in addressing this pain by becoming an activist, working with NGOs on the FGM/C issue before migration, thus modifying pain into pride [20]. The Pastor-Braco et al. study, which discusses breaking the taboo surrounding the traditional practices, has led to women discussing sexual autonomy in both origin and host communities and was identified as a first sub-category. Some participants see a tangible change and openly oppose the practice [18]. Similarly, sub-Saharan women in Canada followed a similar pattern. In this second sub-category, besides actively challenging the taboo on FGM/C, participants engaged in conversations with other mothers, fostering positive emotions, mutual understanding, and a shared commitment to protect their daughters [33]. In both studies, mothers’ collective resistance created a new social network of solidarity, aiding their adaptation to the post-migration cultural setting.

### 4.8. Limitations

This scoping review has several limitations. Firstly, while it synthesizes a wide array of research studies, most of the selected studies have small sample sizes, limiting the generalizability of findings on attitude changes towards FGM/C to post-migration contexts. Secondly, inconsistencies in the design and methodology of identified studies, including insufficient explanation of qualitative studies’ researcher roles and bias mitigation, and the absence of measures for non-responders in quantitative studies, introduce potential biases into the review analysis. Additionally, the review is constrained by a 10-year timeframe, with publications only updated until 2012, potentially overlooking relevant literature addressing the research question. Thirdly, some studies focused on specific subsets of migrants, such as anti-FGM/C activists or mothers raising uncut girls, possibly introducing selection bias towards positive attitudes regarding discontinuing the practice. Given that FGM/C is illegal in Western countries, there might be response and social desirability biases due to fear of legal repercussions. However, some participants in studies expressed support for the practice in the post-migration context, regardless of current anti-FGM/C legislation [17,18,21]. While migration and cultural change are associated with attitude changes towards FGM/C, it is important to note that attitudinal changes can occur before migration [30]. Therefore, the definitive role of migrating to Western countries in altering attitudes across all migrant communities cannot be conclusively assessed. Finally, some studies were excluded due to unclear links between migration and attitude changes towards discontinuing the practice, and full texts of certain studies were inaccessible. Despite efforts to minimize selection bias through consensus among reviewers, there may be other factors related to the research question that were not considered in this analysis.

## 5. Conclusions

This comprehensive review underscores the multifaceted factors influencing attitudes and behavior toward FGM/C among migrating populations in Western countries. These factors interact across different levels, particularly in environments where opposing views on FGM/C exist, where the practice is illegal, and social pressure to continue it is diminished, prompting individuals to adopt new social norms. While laws alone may not suffice, they can complement culturally sensitive prevention strategies and health promotion policies. Educational opportunities for migrants, including men and women, are crucial, providing them with agency for autonomy and the ability to envision alternatives to FGM/C. Young girls and boys should also be educated and included in programs alongside their parents, giving them the agency to reject the practice, seek assistance, and understand their rights. Men play an active role in promoting their daughters’ health, often serving as primary decision-makers in the family. Future research can utilize these findings to extend beyond FGM/C, particularly for younger individuals of the second and third generations, and design programs focused on sexual and reproductive health. These programs can address various cultural themes related to virginity, pleasure, and sexual scripts.

## Figures and Tables

**Figure 1 ijerph-21-00528-f001:**
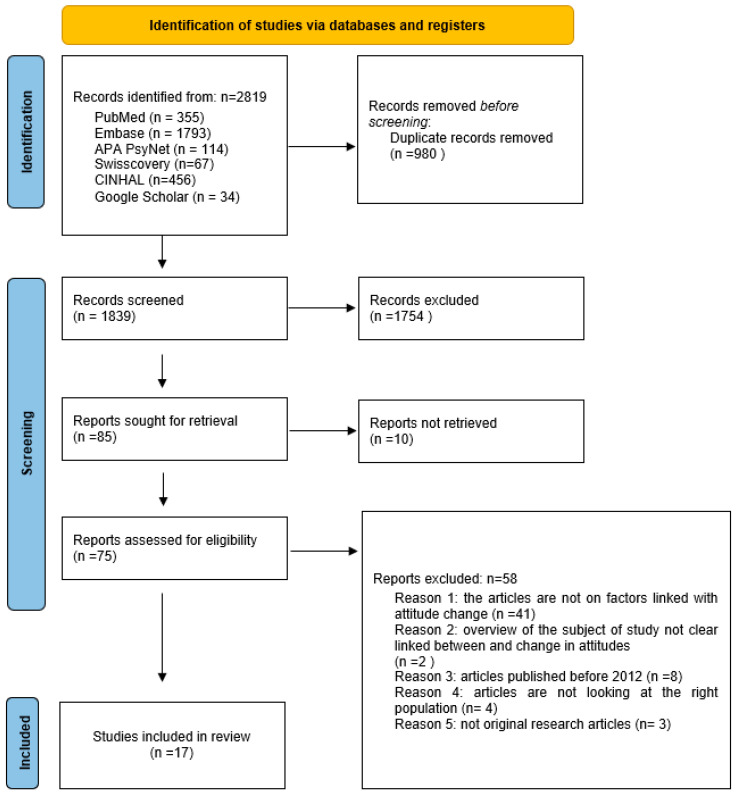
Prisma Flow Diagram.

**Table 1 ijerph-21-00528-t001:** MeSH words used in the literature search.

Concept 1 Population	Concept 2 Attitude	Concept 3 FGM/C	Concept 4 Western Countries
Female Women Transients AND Migrants Emigrants AND Immigrants Male Adult Men	Attitude Knowledge Culture Acculturation Behavior Intention Perception	circumcision, female	Europe France Germany Italy Netherlands Norway Sweden Spain Switzerland United Kingdom North America United States Canada Australia New Zealand Israel

**Table 2 ijerph-21-00528-t002:** Characteristics of studies included in the review (N = 17).

Author, Year and Reference	Study Type	Country	Study Quality	Population, Country of Origin and Current Setting	Aim	Factors Identified as Associated with Change in Attitudes and Behavior FGM/C
Agboli et al. (2020) [30]	Qualitative study Interview Life-story narrative approach	Belgium	18/20 high	N = 15 Female, from FGM/C-practicing community in East or West Africa. Living in Belgium	Aim: to identify and describe turning points that have been defined as significant and critical events in the lives of the women, and which have engendered change in their attitudes towards the practice of FGM/C	- encounters with health professionals that bring awareness and information about anatomy of the vulva - education about anatomy and sexuality as well as the health consequence of the practice - social interactions with other cultures unable to -question “normality” and their own culture that promotes FGM/C - experiences of motherhood and the urge to protect their daughters - repeated pain during sexual or reproductive activity - witnessing the effects of some harmful consequences of FGM/C on loved ones
Akinsulure-Smith et al. (2017) [24]	Quantitative cross-sectional study audio computer-assisted self-interview and self-administered questionnaire	United States of America	30/40 moderate	N = 107 Female and male, from Sierra Leone, Guinea, Gambia, and Mali. Living in New York City, USA	Aim: to explore knowledge and attitudes toward female genital mutilation (FMG/C) in West African male immigrants in New York City	- increased health problems (HIV transmission, urinating, childbirth and fertility issues) - knowledge of the practice is illegal - increased sexual problems - human rights - no preference or higher preference for uncut women for dating and marriage
Catania et al. (2016) [29]	Qualitative Focus groups	Italy	18/20 high	N = 50 Male, from Somalia, Eritrea, Ethiopia, Benin, Egypt and Nigeria. Living in Italy	Aim: to investigate the attitudes, knowledge and beliefs regarding female genital mutilation/cutting (FGM/C) of six groups of immigrant men from countries where FGM/C is practiced and to identify their role in the decision-making process of circumcising their daughters	- awareness of the women’s health dangers main reason to reject practice (through awareness and information campaigns) - Length of years spent in the host country(>10 years), level of education and having a job increases rejection of the practice - knowledge gain after migration changed attitudes - older and educated men tend to have negative attitudes towards FGM/C - not a religious requirement - practice does not control female sexuality - negative effect on men and on the couple’s sexual life - men being involved in the decision making
Gele et al. (2012) [17]	Qualitative Focus groups	Norway	20/20 high	N = 38 Female and male, from Somalia. Living in Norway	Aim: to explore the attitudes of Somalis living in Oslo, Norway, toward the practice of FGM/C	- education about anatomy and sexuality as well as the health consequence of the practice - not a religious requirement - perception that FGM/C erodes females’ sexual feelings - social environment in host country supports FGM/C discontinuation - change in perceptions toward uncircumcised girls (better marriage perspective)
Gele et al. (2012) [21]	Quantitative cross-sectional study	Norway	35/40 high	N = 214 Female, and male, from Somalia. Living in Norway	Aim: to investigate whether or not Somali immigrants’ attitudes toward the practice has improved in favor of its abandonment	- length of years spent in the host country (>14 years) - not a religious requirement - lower pressure to pursue the practice - gender difference (women less supportive) - change in perceptions toward uncircumcised girls
Gele et al. (2015) [27]	Qualitative Interviews	Norway	17/20 high	N = 24 Female and male, from Somalia. Living in Norway	Aim: to explore the attitudes toward FMG/C among young Somalis between the ages of 16 and 22 living in the Oslo and Akershus regions of Norway	- social environment in the host country that rejects the practice - FGM/C is unethical and a violation of human rights - criminalization of FGM/C in Norway - better perspective of marriage for uncut girl
Hassanen et al. (2019) [32]	Qualitative In-depth unstructured and semi-structured interviews	Australia	18/20 high	N = 50 Female and male, migrants from the Horn of Africa. Living in Melbourne, Australia	Aim: to examine the effects of migration on the practice and perception of Female Genital Mutilation or Cutting (FGM/C) among Horn of Africa immigrants in Melbourne, Australia	- gender view of the practice as harmful and change in gender dynamics - immigration to Australia to change their perceptions about the practice - easy access to different ideas, norms, values, knowledge and information about the implication of the practice - speaking freely about the practice - legal consequence of engaging in the practice
Kawous et al. (2022) [28]	Qualitative Semi-structured focus group discussions	Netherland	19/20 high	N = 55 Female and male, from Somalia, Guinea, Ethiopia, Eritrea, Sudan, Egypt, Togo, Ghana, Sierra Leone, and Iraq. Living in the Netherlands	Aim: to explore the attitude and intention of migrant populations in the Netherlands towards FGM/C, among the newly arrived	- adverse health consequences - not a religious requirement - change in behavior and in the discourse on FGM/C (breaking the taboo) - different cultural significance of FGM/C between host country and country of origin) - better perspective of marriage for uncut girl - perceived behavioral control (resist social pressure from people)
Koukoui et al. (2017) [33]	Qualitative Semi-structured interviews	Canada	19/20 high	N = 15 Female, from Ivory Coast, Somalia, Djibouti and Ethiopia, West Africa, Mali, Guinea and Egypt. Living in Canada	Aim: to shed light on mothers’ perceptions of the meaning and cultural significance of the practice and to gain insight into their mothering experience of ‘uncut’ girls	- not a religious requirement - avoiding negative health consequence for daughters and intimate unsatisfaction - men’s refusal to cut their daughters - advice from health professional (pediatrician) - personal experience of negative outcome of FGM/C - legal repercussion (Youth Protection) - perceived men’s preference for uncut girls - avoid future conjugal difficulties for daughters - avoid harmful and dangerous effect of FGM/C
Lien et Schultz (2013) [20]	Qualitative Interviews	Norway	16/20 high	N = 26 Female, from Eritrea, Ethiopia, Gambia, Ghana, Kenya, and Somalia	Aim: to describe and analyze the way in which persons who were socialized in a cultural context where FGM/C is highly valued receive and process information that contradicts and devalues the meaningful norms and traditions, they internalized as children	- education: obtaining information about the health risks (anatomical classes, videos, slides, etc.) - psychological, emotional pain of the memory of the cutting - not a religious requirement - concrete experience of working at a hospital: seeing other women with FGM/C - attending seminars and conferences about the practice - discussions with family and friends about FGM/C - meeting people with the same background that oppose the practice
Martínez-Linares et al. (2022) [25]	Qualitative Interview	Spain	18/20 high	N = 12 Female, from Gambia, Senegal, Nigeria, Mali, Burkina Faso, Equatorial Guinea, and Guinea. Living in Spain	Aim: to describe and understand the lived experiences and opinions of sub-Saharan women living in Spain in relation to female genital mutilation	- avoiding negative health consequence - critically reflecting on what the experience of genital mutilation meant for a woman - preventing the suffering of their daughters - legal consequence of engaging in the practice
Pastor-Bravo et al. (2021) [18]	Qualitative Life stories telling and open interview	Spain	20/20 high	N = 24 Female, from Senegal, Gambia and Nigeria. Living in Spain	Aim: to find out the elements that support the continuation of FGM/C and those that promote the change in attitudes and fight against FGM/C from the perspective of the sub-Saharan women themselves who reside in Spain	- knowledge about health consequences and close negative experiences (informed about the health consequences) - institutional support to oppose FGM/C (including awareness of rights and their violation, knowledge of the law, supporting legislation, importance of awareness raising and effect of law in their home countries)
Shahawy et al. (2019) [19]	Qualitative Semi-structured interview	United States of America	18/20 high	N = 42 Female and male, from Egypt, Eritrea, Ethiopia, Ghana, Mali, Nigeria, Somalia, Sudan, and the United Arab Emirates. Living in Boston/USA	Aim: to document the FGM/C-related perceptions and experiences of immigrant women and men in Boston, Massachusetts, and to attempt to understand the effect of migration on these perceptions	- negative views of FGM/C in Boston (which can led to stigma) - men’s preference for women without FGM/C and supposedly who have stronger sexual desires - men’s role in advocating for change and using their dominant position - change in women’s views towards the practice - involving religious leadership to denounce the practice - migration as it broadens the social network and new social networks and shifts to the reference group regarding FGM/C - generational differences: the practice is seen as backwards practice - emigration and migration that has an effect on their personal view and at the community level
Shahid, et Rane (2017) [31]	Quantitative cross-sectional study	Australia	37/40 high	N = 67 Male, from Somalia, Nigeria Kenya, South Sudan, Liberia, Egypt, Ethiopia. Living in Australia	Aim: to elicit the poorly understood perceptions that young, sub-Saharan African, migrant males residing in Townsville, Australia, have of FGM/C	- level of education - length of years spent in the host country - knowledge about the negative health consequence - knowledge of one’s health complications due to the negative outcome of the practice
Vogt et al. (2017) [26]	Quantitative test computerized	Switzerland and Sudan	37/40 high	N = 2344 Female and male, Sudanese and Swiss Sudanese. Living in Sudan and Switzerland	Aim: to compare attitudes about female genital cutting among Sudanese living in Switzerland with attitudes among Sudanese living in Sudan	- risk for girls to be cut declines as implicit attitudes toward uncut girls become more favorable - length of years spent in the host country - not religious requirement - the selective migration and acculturation impact factor, the local norms, the threat of legal sanctions, social pressure - mothers’ education - personal attitude towards FGM/C
Wahlberg et al. (2017) [23]	Quantitative cross-sectional study	Sweden	37/40 high	N = 372 Female and male, from Somalia. Living in Sweden	Aim: to present the primary outcomes from a baseline study on attitudes towards female genital cutting (FMG/C) after migration	- length of time spend in host country (≥4 years) - higher support for choosing this among newly arrived - low support for continuation of practices causing anatomical change
Wahlberg et al. (2019) [22]	Quantitative cross-sectional study	Sweden	39/40 high	N = 648 Female and male, from Somalia. Living in Sweden	Aim: to investigate correlations between the Somali Swedish attitudes towards female genital cutting (FMG/C) and their perceptions about other Swedish Somalis’ attitudes	- perceived group opinion of FGM/C affects individual woman’s attitude towards FGM/C - support of FGC correlated with perception about what the group prefer - change in preference to marry women with no FGM/C - reduction in pressure in host country where FGM/C is not the norm and there is legislation against - new social networks and shift to the reference group regarding FGM/C

**Table 3 ijerph-21-00528-t003:** Seven factors including sub-categories identified as associated with change in attitudes and behavior towards FGM/C.

Level	Factors	Sub-Category
Social	Legislation and law against FGM/C in host countries and legal repercussion	- discourse opposing FGM/C within the host country
Community	2. FGM/C is not mandated by religion; it is a manifestation of traditional culture rather than a religious obligation	- involving religious leaders
	3. Enhancing education regarding the practice	- national campaign and medical personnel informing about the practice - combining knowledge to raise awareness
	4. Effect of migration and cultural change	- duration of residency in the host country and subjective norms - reduced social pressure for FGM/C in host countries
Interpersonal	5. Awareness of the negative health consequences of FGM/C	- increased sexual and reproductive health problems - reshaping motherhood and fatherhood to protect their daughters from FGM/C
	6. Change in perception towards uncircumcised girls and better or same perspective of marriage for uncut girls	
Personal	7. Individual disposition to oppose FGM/C and sense of self-agency.	- change in behavior and in the discourse of FGM/C - developing supporting networks

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
