# Peer review of "Investigating Factors Associated with Migration and Cultural Adaptation in Relation to Change in Attitudes and Behavior towards Female Genital Mutilation/Cutting (FGM/C) among Populations from FGM/C-Practicing Countries Living in Western Countries: A Scoping Review"

_ijerph, 2024, doi:10.3390/ijerph21050528_

Round 1

Reviewer 1 Report

Comments and Suggestions for Authors

Thank you for the opportunity to review this well written scoping review. The findings and recommendations are relevant for targeted interventions across the multi level factors. Key insights highlight pre and post migration factors associated with FGM/C attitudes/factors associated with supporting or not supporting the practice.

A strengh of the review was inclusion of multiple methods over multiple countries that include cultures who undertake FGM/C. Excellent gender analysis and inclusion of migrant men and the connection between education and adoption of the practice. Future research should consider migrant men and their role in health promotion as this is a gap in the literature.

Table 3 is clear showing the 7 factors and summary of the findings. Discussion begins with 4.1. top of p.2 of 22 line 319 but not clear if these are subcategories or themes from the findings that follow. Example: is 4.1. Awareness of the negative health consequences of FGM/C 320 321 4.1.1. Increased sexual and reproductive health problem with FGM/C (a sub category?)

The Social 1. Legislation and law against FGM/C in host countries and legal repercussion -discourse opposing FGM/C within the host country- is the first factor and subcategory in Table 3 then the discussion starts with  4.1. Awareness of the negative health consequences of FGM/C (that is a bit unclear) in terms of threading the ideas; howeve the content is clear and consistent with the findings.

The conclusion brings awareness of policy and implications across multi level factors and targeted interventions- very clear and informative for health providers and policy makers. 

May want to include future directions for research.

Thank you

Author Response

We sincerely appreciate the time you dedicated to reviewing our article. Thank you for your valuable feedback on our manuscript. Please see the detailed feedback below and review the attached manuscript with track changes.

Comments 1

Discussion begins with 4.1. top of p.2 of 22 line 319 but not clear if these are subcategories or themes from the findings that follow. Example: is 4.1. Awareness of the negative health consequences of FGM/C 320 321 4.1.1. Increased sexual and reproductive health problem with FGM/C (a sub category?) The Social 1. Legislation and law against FGM/C in host countries and legal repercussion -discourse opposing FGM/C within the host country- is the first factor and subcategory in Table 3 then the discussion starts with 4.1.

Response 1

In response to your comments, we have adjusted the structure of the discussion section. The factors and subcategories are now ordered according to Table 3, starting from line 310. The discussion commences with the first factor listed in Table 3, followed by the successive factors as outlined below. At the beginning of each of the 7 sections, we have included a sentence that specifies the factor. Within the text, we have also mentioned the associated sub-category if applicable.

Lines 310-340: 4.1 Legislation and law against FGM/C in host countries and legal repercussions -discourse opposing FGM/C within the host country.

sub-category: societal disapproval of FGM/C in Western countries

Lines 342-358: 4.2.FGM/C is not mandated by religion; it is a manifestation of traditional culture rather than a religious obligation

Lines 360-384:  4.3. Enhancing education regarding the harm of FGM/C and increased knowledge of human anatomy.

2 sub-categories: community education and awareness-raising

Lines 387-410: 4.4. Effect of migration and acculturation

sub-categories: longer duration of residence and reduced social pressure

Lines 412-438: 4.5. Awareness of the negative health consequences of FGM/C

sub-categories: 1) increased concerns about sexual and reproductive health issues related to FGM/C, and 2) a shift in parental roles aimed at protecting daughters from FGM/C.

Lines 440-452: 4.6. Change in perception towards uncircumcised girls and better or same perceptive of marriage for uncut girls

Lines: 454-476 4.7. Individual disposition to oppose FGM/C

sub-categories: breaking the taboo and creating a new social network.

Comments 2: include future directions for research.

Response 2

Thank you for your comment we have addressed this aspect in line 514-518: "Future research can utilize these findings to extend beyond FGM/C, particularly for younger individuals of the second and third generations, and design programs focused on sexual and reproductive health. These programs can address various cultural themes related to virginity, pleasure, and sexual scripts."

Reviewer 2 Report

Comments and Suggestions for Authors

The article presents the results of a literature review with the aim of exploring the effects of migration and acculturation on stopping female genital mutilation/cutting (FGM/C) in migrant communities.

The potential of the proposal could be strengthened by a critical reflection on the concept of acculturation - historically known as a Eurocentric concept criticized for diminishing certain people while they would imitate their counterparts of the target ‘superior’ culture. In particular, the article needs to dialogue a lot more with the critical literature on the concept of acculturation.

The proposal uses a second concept, which in turn requires greater attention and possibly a certain degree of prudence - ‘empowerment’. Also in this case, the article would benefit dialoguing with the critical literature on this concept, which emphasizes its limits, from the substantial negligence of structural obstacles to gender equality.

From a methodological point of view, one can see that the publications indexed in some of the main international databases was not taken into consideration. This would require at least a more careful explanation of why including certain databases and not others.

Comments on the Quality of English Language

Minor editing of English language required.

Author Response

We sincerely appreciate the time you dedicated to reviewing our article. Thank you for your valuable feedback on our manuscript. Please see the detailed feedback below and review the attached manuscript with track changes.

Comments 1

The potential of the proposal could be strengthened by a critical reflection on the concept of acculturation - historically known as a Eurocentric concept criticized for diminishing certain people while they would imitate their counterparts of the target ‘superior’ culture. In particular, the article needs to dialogue a lot more with the critical literature on the concept of acculturation.

Response 1: Lines 49 to 85

Based on your feedback, we've revised the entire manuscript regarding the concept of acculturation. We've removed the term "acculturation" as a simple description of cultural change or adaptation. Instead, we now describe it as the complex process by which migrants integrate into a new society, gradually learning its social and cultural norms. We've opted to use terms like "cultural change" or "cultural adaptation" to better capture the nuanced nature of this process. We've highlighted the importance of facing social, economic, and discriminatory challenges that can significantly affect this adaptation to a new environment. In the existing literature on FGM/C-affected populations, social adaptation is typically assessed through various factors. However, in this review, we've aimed to understand the interplay of these factors and how they drive change in first-generation migrants. We've also included articles that demonstrate the multifaceted nature of social adaptation in the introduction and the discussion, emphasizing that it's a complex process that requires time, culturally sensitive approaches, and awareness-raising efforts within both the migrant communities and host countries. We stress that cultural change is dynamic, involving mutual influences between migrants and the host society.

Comments 2

The proposal uses a second concept, which in turn requires greater attention and possibly a certain degree of prudence - ‘empowerment’. Also in this case, the article would benefit dialoguing with the critical literature on this concept, which emphasizes its limits, from the substantial negligence of structural obstacles to gender equality.

Response 2: Lines 454-476

We have revised the text by eliminating the term "empowerment" as we have found that other concepts better suit this context. Specifically, in factor 7, we have substituted this term with "sense of self-agency". Our findings indicate that fostering self-agency among both men and women can mitigate social conformism to the practice, and we are approaching the topic from this perspective. An example is in the conclusion section «Educational opportunities for migrants, including men and women, are crucial, providing them with agency for autonomy and the ability to envision alternatives to FGM/C. Young girls and boys should also be educated and included in programs alongside their parents, giving them the agency to reject the practice, seek assistance, and understand their rights. “

Comments 3

From a methodological point of view, one can see that the publications indexed in some of the main international databases was not taken into consideration. This would require at least a more careful explanation of why including certain databases and not others.

Response 3: Lines 127-134

It is unclear which databases you were referring to that were not taken into consideration. The selected databases were made according to their relevance as sources for medical, behavioral, and healthcare research. For instance, PubMed contains millions of citations for biomedical literature from MEDLINE, life science journals, and online books. Furthermore, we utilized Embase, which offers insights through its structured full-text indexing, unlike Scopus searches that primarily focus on abstracts and citations. As Scopus does not utilize Emtree indexing it may retrieve considerably fewer results than Embase. To the best of our knowledge, these selected databases hold the potential to encompass the relevant literature without encountering excessive interference. Our selection process yielded over 900 duplicate results across the databases, indicating that we have achieved data saturation.

Reviewer 3 Report

Comments and Suggestions for Authors

Review

Dear authors, congratulation for this manuscript, the topic being of great interest.

However, I have a few remarks:

After reading the manuscript, I wonder what the main question is?

The introduction contains relevant elements from the literature, and the references corresponds to the topic.But the results obtained and presented in table 2 contain overlaping data, which I find difficult to follow. Especially taking into account that in some studies included the sample size is small.

In the discussion chapter there are too mny subchapters, for example 4.4.1 and 4.4.2 address the same problem.Maybe the authors manage to merge them.

In the end, the conclusions are too long, they should include some take-home messages.

Author Response

We sincerely appreciate the time you dedicated to reviewing our article. Thank you for your valuable feedback on our manuscript. Please see the detailed feedback below and review the attached manuscript with track changes.

Comments  1

After reading the manuscript, I wonder what the main question is?

Response 1: Lines 86-89

In response to your feedback, we have included our research question in the final section of the introduction, thereby providing clarity on the focus of this scoping review and contextualizing its scope.

Comments 2

The results obtained and presented in Table 2 contain overlapping data, which I find difficult to follow. Especially taking into account that in some studies included the sample size is small.

Response 2

Table 2 exhibits the variables pertaining to shifts in attitudes as documented in the referenced studies. These selected factors were identified in alignment with the research question, with the aim of pinpointing variables linked to attitude changes within the post-migration scenario. Consistent terminology was utilized across the tables to illustrate that common factors were identified across multiple articles, enhancing clarity regarding the elements retained for analysis. The resemblance among various articles, highlighting similar influential factors in attitude change, is corroborated by the ecological model of behavior change. This model proposes that change occurs across different dimensions, with interconnected factors playing a role. Factors such as potentials legal repercussions, absence of religious mandates, duration of stay in the host country, reduced social pressure, and increased awareness of the negative health consequences are associated with diminished support for FGM/C in the post-migration context, as evidenced by the results. Additionally, we have noted in the limitations of the study that some of the selected articles had small sample sizes, which restricts the generalizability of findings regarding attitude changes towards FGM/C in post-migration contexts.

Comments 3

In the discussion chapter there are too mny subchapters, for example 4.4.1 and 4.4.2 address the same problem.

Response 3: Line 310-476

In light of your comments, we have restructured the discussion section to align with Table 3. Furthermore, we have condensed the length of the discussion to enhance clarity and facilitate easier comprehension of the various arguments presented within this section.

Comments 4

In the end, the conclusions are too long, they should include some take-home messages.

Response 4 : Line 503-518

Following your feedback, we have clarified the overarching message of the entire scoping review and provided guidance on utilizing the findings to strengthen further research or inform policies aimed at addressing this topic.